# Novel Ex Vivo DOAC Removal Methods Reduce Interference in Lupus Anticoagulant Testing

**DOI:** 10.3390/diagnostics12102520

**Published:** 2022-10-17

**Authors:** Paula Savola, Marja Lemponen, Lotta Joutsi-Korhonen, Tuukka A. Helin

**Affiliations:** Department of Clinical Chemistry, HUS Diagnostic Center, Helsinki University Hospital and University of Helsinki, 00290 Helsinki, Finland

**Keywords:** active charcoal, direct anticoagulant, lupus anticoagulant, DOAC, DOAC Filter^®^, DOAC-Stop™

## Abstract

Direct oral anticoagulants (DOAC) interfere in laboratory coagulation testing. The aim here was to study how commercial DOAC removal methods, DOAC Filter^®^ and DOAC-Stop™, perform to eliminate DOAC concentrations and false positive results in lupus anticoagulant (LAC) testing. We acquired 50 patient samples with high concentrations of DOACs: apixaban (n = 18, range 68–572 ng/mL), dabigatran (n = 8, range 47–154 ng/mL), edoxaban (n = 8, range 35–580 ng/mL) and rivaroxaban (n = 16, range 69–285 ng/mL). DOACs were removed ex vivo with either DOAC Filter^®^ (n = 28) or DOAC-Stop™ (n = 22). Additionally, commercial control and calibrator samples were studied (n = 13 for DOAC Filter^®^, n = 14 for DOAC-Stop™). LAC screening was performed before and after DOAC removal. Both DOAC Filter^®^ and DOAC-Stop™ were effective in removing DOAC concentrations in samples: DOAC concentrations decreased to median of 0 ng/mL (range 0–48 ng/mL). Only one sample had more than residual 25 ng/mL of DOAC (apixaban). Before DOAC removal, 96% (48/50) of patient samples and over 90% (12/13 DOAC Filter^®^, 13/14 DOAC-Stop™) of control/calibrator samples were positive in the LAC screening. In patient samples, LAC screening turned negative in 61% (17/28) after DOAC Filter^®^ and 45% (10/22) after DOAC-Stop™ treatment. All control samples became negative after DOAC removal. In conclusion, DOAC removal ex vivo reduces false positives in LAC screening. DOAC removal halved the need for confirmation or mixing tests- Although a subset of patients would require further testing, DOAC removal reduces unnecessary repeated LAC testing.

## 1. Introduction

Common thrombophilia workup schemes include testing for lupus anticoagulant (LAC). However, frequently used anticoagulation therapy may cause false positive results in coagulation-based assays, which results in repeated testing. Direct oral anticoagulants (DOACs) are relatively novel agents which inhibit coagulation factor Xa (rivaroxaban, apixaban, edoxaban) or thrombin (Factor IIa, dabigatran) directly. The ease of use and lack of routine monitoring requirements have increased the rates of using these drugs over the more traditional vitamin K antagonists (VKAs) [1]. DOACs interfere often with coagulation-time-based diagnostic assays. In patients receiving DOAC treatment, approximately 50–80% of LAC tests may be positive [2]. Thus, elimination of the effect of anticoagulants in the tested plasma ex vivo would be beneficial to improve the reliability of testing.

Recently, two commercially available methods to eliminate DOAC effect have been introduced. DOAC-Stop™ is a novel, activated-charcoal-based compound, which eliminates effects of DOAC. It has been used to remove DOAC prior to LAC testing [3]. DOAC-Stop™ reduces apixaban, dabigatran, and rivaroxaban concentrations in spiked plasma samples to values below detection limits [4,5], but it may not remove completely the DOAC effect on patient samples in all cases [6]. Recently, Tripodi and coworkers demonstrated that DOAC-Stop™ is very effective in reducing DOAC concentrations ex vivo in patient plasma samples, although the remaining concentrations of edoxaban were slightly higher than those of apixaban, dabigatran, and rivaroxaban [7]. Results from other active charcoal-based products have also been published [8,9].

DOAC Filter^®^ is a novel type of laboratory device which eliminates DOAC from sample plasma with light centrifugation via a DOAC-adsorbing filter [10]. Here, we aimed to assess the value of both DOAC Filter^®^ and DOAC-Stop™ in a real-life clinical laboratory setting by performing repeated LAC screening by RVVT (Russell’s viper venom) and APTT (activated partial thromboplastin time) testing before and after DOAC removal.

## 2. Materials and Methods

Surplus plasma samples (n = 50) from patients receiving DOAC treatment were collected for our study; these samples were referred to our laboratory for either DOAC concentration measurement (n = 44), thrombophilia testing (n = 4) or other coagulation test (n = 2). Patient blood samples were collected to tubes containing 109 mM (3.2%) sodium citrate (BD Vacuette, Becton Dickinson, Franklin Lakes, NJ, USA) and handled according to our laboratory protocol, i.e., maximum 2 h at room temperature before separating plasma by centrifugation (2500× *g*, 15 min). If required, the citrated plasmas were frozen (−20 °C). DOAC concentrations were measured by anti-Xa based or diluted thrombin time-based assays (Appendix A). Only samples with a DOAC concentration of more than 30 ng/mL were included in sample selection, and only 3/50 samples had a DOAC concentration less than 50 ng/mL (range 35–47 ng/mL). Samples were anonymized. Additionally, commercially available lyophilized DOAC controls and calibrators in plasma matrix were used (n = 13 for DOAC Filter^®^ treatment and n = 14 for DOAC-Stop™ treatment, Appendix A).

DOAC concentration measurements and LAC testing, i.e., APTT and RVVT tests, were performed before and after DOAC-Stop™ (Haemochrom, Essen, Germany) or DOAC-Filter^®^ (Diagnostica Stago, Paris, France) treatments. DOAC removal ex vivo protocols were performed according to manufacturers’ instructions. In DOAC-Stop™ treatment, one tablet (18 mg) was dispensed to 1 mL of patient plasma, followed by centrifugation at 2500× *g* for 5 min and supernatant collection. Edoxaban and dabigatran samples were centrifuged twice to obtain clear plasma because of black residual particles of the charcoal product. In DOAC-Filter^®^ treatment, samples were gently centrifuged through a filter (300× *g* 15 min). Due to sample amount constraints, we performed only screening and confirmation tests for LAC testing and no mixing tests were performed. Therefore, we interpreted all samples with a positive screening test as “LAC-positive”. DOAC-Filter^®^ was a kind gift from Diagnostica Stago. DOAC-Stop™ was purchased from Haemochrom.

All coagulation tests were performed in our accredited clinical laboratory (Helsinki University Hospital), using the BCS-XP analyzer (Siemens Healthineers, Erlangen, Germany) for LAC testing and ACL (Werfen, Bedford, MA, USA) for other assays. LAC testing was performed with two different screening and confirmation assays: RVVT (screening assay LA1 and confirmation assay LA2) and APTT (screening assay FSL and confirmation assay FS), all from Siemens Healthineers (Erlangen, Germany). Reagents are listed in Appendix A.

In LAC screening testing, we classified samples as positive with an RVVT-based screening test result equal or over 42 s and with APTT-based screening equal or over 32 s. A sample was considered positive in the LAC screening test if either of the screening tests was positive. If the sample volume was insufficient to determine both screening tests after treatment with DOAC Filter^®^ or DOAC-Stop™, the overall screening test was considered negative if the performed test (either RVVT or APTT) was negative. The sample volume was insufficient for 10 samples for all screening and confirmation tests after DOAC Filter^®^ treatment. In two samples, only RVVT LA2, APTT FS, and APTT FSL tests were performed. In two other samples, only RVVT LA1, APTT FSL and APTT FS tests were performed. In six samples only RVVT LA1, RVVT LA2, and APTT FS tests were performed. Additionally, for two edoxaban-containing samples, sample volume was insufficient to determine edoxaban concentration after DOAC Filter^®^ treatment.

Due to sample volume constraints, we did not perform mixing studies. However, in our clinical laboratory, a ratio of RVVT screening and confirmation test results equal or less than 1.25 requires mixing studies before reporting the final result. A ratio >1.25 is considered positive without mixing studies. Similarly, a ratio of APTT screening and confirmation test results equal or less than 1.1 require mixing studies before reporting the final result, but a ratio >1.1 is reported as positive.

DOAC concentrations were determined based on their anti-Xa (HemosIL Liquid Anti-Xa, Werfen, Bedford, MA, USA) or diluted thrombin time activity (HemosIL Direct Thrombin Inhibitor Assay, Werfen, Bedford, MA, USA) with appropriate calibrators, all by ACL analysers (Appendix A). In addition, we analyzed four samples from patients who did not use DOACs for thrombin time, prothrombin time, APTT, protein C activity, antithrombin activity, protein S free antigen, factor VIII activity, activated protein C activity and LAC. Reagents and analyzers are listed in Appendix A. Antithrombin, protein C and factor VIII activity were measured with clotting-based methods. Protein S free antigen was measured with a latex particle agglutination assay.

The primary outcome was the frequency of APTT and RVVT test results turning from positive to negative after DOAC removal (a binary variable). All statistical analyses were performed with R (version 3.6). [11] All tests performed were nonparametric unless stated otherwise. Spearman correlation, the McNemar test, and the paired sign test were used, when appropriate.

This study was approved by the HUS Diagnostic Center institutional Review Board (no. 10/2022, approval date 7 March 2022).

## 3. Results

### 3.1. Effects of DOAC Removal Treatment on DOAC Concentrations and Coagulation Tests

DOAC removal decreased DOAC concentrations in all the samples: concentrations decreased to a median concentration of 0 ng/mL with DOAC Filter^®^ and 1 ng/mL with DOAC-Stop (Table 1, Figure 1).

One sample had a concentration of apixaban 48 ng/mL after DOAC Filter^®^ treatment. We also analyzed a limited sample of four patient samples without DOAC therapy to assess if DOAC-Stop™ or DOAC Filter^®^ had effects on coagulation tests. Measured tests, such as thrombin time, prothrombin time, APTT, and protein C activity did not alter in the context of the expected performance of the assay (Appendix A).

### 3.2. DOAC Concentration Correlates with RVVT-Based Coagulation Time

Of all patient samples analyzed, 96% (48/50) had a positive LAC screening test before DOAC removal: 96% (48/50) had a positive RVVT screening test result and 58% (29/50) had a positive APTT screening test. LAC testing results obtained before DOAC removal show that the RVVT-based LA1 screening test coagulation time seems to prolong with DOAC concentration, especially with apixaban and rivaroxaban samples (Figure 2, Spearman correlation *p* < 0.001). For APTT-based screening testing, the concentration of the DOAC did not associate as consistently with prolonged coagulation times (Appendix A): only patient samples containing rivaroxaban showed a significant correlation between concentration and the FSL clotting time (Spearman r = 0.8 *p* = 0.0002). Calibrator and control samples showed also highly positive LAC screening test results, RVVT-based coagulation screenings seemed to be more often prolonged (93%; 13/14 DOAC-Stop™ and 92%; 12/13 DOAC Filter^®^) than those of APTT-based assays (79%; 11/14 DOAC-Stop™, 69%; 9/13 DOAC Filter^®^).

In concordance with efficient removal of DOAC concentrations in patient and control samples (Figure 1), the LA1 screening test clotting time shortened with all DOACs for both DOAC Filter^®^ and DOAC-Stop™-treated samples (Figure 3). FSL screening test results did not shorten significantly in apixaban or edoxaban samples, but they did with dabigatran (Figure 4, *p* = 0.016) and rivaroxaban (Figure 4, *p* = 0.039). Confirmation tests showed similar results as screening tests (Appendix A): RVVT-based confirmation test (LA2) clotting times were reduced by both DOAC Filter^®^ and Stop™ treatments with all DOACs (Appendix A). APTT-based (FS) confirmation test clotting time was shortened by both DOAC Filter^®^ and DOAC-Stop™ treatments in dabigatran and edoxaban samples (Appendix A). Thus, DOAC removal ex vivo reduces false positives in LAC assays.

In clinical context, the most important parameter is the elimination of false positives in LAC testing. After treatment with DOAC Filter^®^, RVVT-based positive screening test results declined to 18% (7/39) from 95% (39/41; Appendix A; *p* < 0.0001). APTT-based screening test positivity declined to 17% (6/35) from 61% (25/41; Appendix A; *p* = 0.0008). Similarly, after treatment with DOAC-Stop™, RVVT-screening test positivity frequency declined to 25% (9/36) from 94% (34/36; Appendix A; *p* < 0.0001) and APTT-screening test positivity to 22% (8/36) from 67% (24/36); Appendix A; *p* = 0.0004). Overall, when combining results from both screening tests, both DOAC-Stop™ and DOAC Filter^®^ reduced false positive screening test results: 73% (30/41) turned negative after DOAC Filter^®^ treatment and 67% (24/36) after DOAC-Stop™ (Appendix A, Appendix A; *p* < 0.0001). Dabigatran samples showed positive RVVT- and APTT-based screening test results in all samples (Appendix A).

### 3.3. Positive Results after DOAC Removal: Real Lupus Anticoagulant?

The patient samples in this study were selected based on suspicion of DOAC interference. However, a subset of patient samples had a positive RVVT or APTT—based LAC screening test result (46%, 23/50) even after treatment with either DOAC Filter^®^ or DOAC-Stop™. However, only 2% (1/50) of patient samples would have been deemed LAC-positive without further confirmation. All control and calibrator samples turned negative after DOAC Filter^®^ or DOAC-Stop™ treatment. One sample was treated with DOAC-Stop™ but had a post-treatment measured concentration of apixaban of 48 ng/mL, others had DOAC concentrations ranging from 0 to 15 ng/mL. Even though these samples remained screening-test-positive after DOAC removal, treatment with DOAC Filter^®^ and DOAC-Stop™ reduced the RVVT-based screening test assay clotting times in these samples (Figure 5A, DOAC Filter^®^ *p* = 0.016, DOAC-Stop™ *p* = 0.0039). APTT-based screening test clotting times did not reduce significantly in these samples (Figure 5B, DOAC Filter^®^ *p* = 0.69, DOAC-Stop™ *p* = 0.73). We were not able to perform mixing tests for these samples due to sample amount constraints, but only one of these patient samples would have been claimed positive (defined by LA1/LA2 ratio > 1.25 or FSL/FS ratio > 1.10) without performing mixing tests. Thus, phospholipid antibody or factor deficiency cannot be confirmed or ruled out in these samples.

## 4. Discussion

We demonstrated that patient samples obtained during DOAC treatment have high rates of positive results in LAC screening tests (96%) and that DOAC concentration in the sample is associated with longer clotting times, especially in RVVT-based screening tests. Elimination of DOAC ex vivo is efficient with both DOAC Filter^®^ and DOAC-Stop™ methodologies, and positive results in APTT- and RVVT-based screening tests are markedly reduced by the treatment.

Our results are in line with previous studies investigating DOAC removal ex vivo in patient samples, in which up to 25–97% of patients have positive LAC screening/test results during DOAC treatment [2,5,6,12]. This far exceeds the prevalence of phospholipid antibodies in the general population [13]. Timing of blood sampling affects the results, because samples drawn at peak drug levels (i.e., 2–3 h after dose) show more laboratory interference than trough-level (12–24 h) samples [7]. Since DOAC concentration correlates with prolonged RVVT and APTT clotting times, the high percentage of positive results in our study is likely due to selecting patient samples with relatively high DOAC concentrations (94% over 50 ng/mL, all over 30 ng/mL). One previous study showed that in follow-up testing with discontinued or withheld DOAC, 75% of initially positive LAC results turn out negative, suggesting that most positive LAC results during DOAC treatment may be false positives [2].

Previous studies have demonstrated the effective removal of DOAC in vitro from samples with active charcoal-based methods for LAC testing [3,4,5,6,7,8,9,10,12,14,15,16,17], but many studies have contained mostly samples spiked with DOAC [3,4,15,16], and most studies have used only DOAC-Stop™ [3,4,5,6,12,14,15,16,17]. To the best of our knowledge, three previous studies have investigated DOAC Filter^®^ [10,18,19]. In contrast, our study has real-life patient samples (with apixaban, dabigatran, edoxaban, and rivaroxaban) as well as commercial calibrator and control samples.

We observed DOAC-induced false positive results—defined as an LAC result turning negative after treatment with active charcoal—mostly in RVVT-based assays. Literature reports that RVVT-based assays are more susceptible to DOAC interference, but especially dabigatran can affect APTT-based assays as well [20,21,22]. Dabigatran dose-dependently prolongs APTT, but its effects are reagent-dependent [22,23]. In dabigatran samples the APTT correlation with dabigatran concentration was not significant in this small study, although the APTT was shortened significantly by DOAC removal. We observed that the APTT-screening test positivity seems to reduce with rivaroxaban and edoxaban as well, so it is likely that some APTT-based assays are affected significantly by various DOACs. As clinical laboratories have slight differences in their APTT tests, our results suggest caution with interpretation with any positive LAC test results during DOAC treatment.

Even after DOAC removal ex vivo, 46% of patient samples showed positive results in LAC screening tests, but all control samples (with spiked DOAC) turned negative. Comparable rates of RVVT screening test positivity (49%) after DOAC-Stop™ treatment have been reported [12]. After confirmation tests, LAC positivity rates have been reported at 27% for DOAC-Stop™ and 14% for DOAC Filter^®^ [18]. RVVT and/or APTT screening test positivity rates after DOAC Filter^®^ treatment may vary according to DOAC and patient population, e.g., Farkh et al. observed that apixaban-treated samples have relatively high post-treatment LAC positivity rates (approximately 50%) [19]. In other studies with patient samples, positive LAC results have disappeared with mixing studies [5], or approximately reduced to 15–20% [6,15] after DOAC-Stop™ treatment. The larger proportion of patient samples with positive screening tests after DOAC removal in our study likely reflects the difference in the use of mixing studies (mixing studies eliminate some false positives) and differences between patient populations and reagents between different studies. Despite this, DOAC removal ex vivo may eliminate the need for repeat testing in most cases, since with DOAC removal, LAC can be excluded in many cases. Repeated sampling and testing might still be required in some cases. Our results highlight that the usefulness of DOAC removal ex vivo should be shown in real patient samples and not only in spiked samples.

In our dataset, only one sample had a significant concentration of DOAC remaining (apixaban 48 ng/mL, positive in the APTT-based screening test). One cannot definitively exclude the effect of another anticoagulant, such as heparin in this sample, because the concentration was measured with an anti-Xa-based assay. Regarding the other samples, we cannot exclude the possibility that they could harbor true in vivo LAC. Since we did not perform mixing studies, factor deficiency remains a possibility and we do not know for certain if these results would have turned out to be negative in further testing. We did not have access to clinical data or medical histories of our patients; thus, we cannot evaluate the pre-test probabilities of real LAC based on clinical context. Additionally, we cannot verify with certainty the DOAC used, and we cannot exclude effects of other anticoagulants in the samples. Thus, ensuring that the laboratory referral is accurate on whether DOAC is used is of primary importance.

DOAC removal strategies have been studied more extensively in LAC testing, but more studies are required to determine if all clotting tests can be interpreted after DOAC removal ex vivo. Our study reviewed different clotting (such as prothrombin time, thrombin time) and thrombophilia (such as protein C activity, antithrombin activity, free protein S) tests before and after DOAC-Stop™ and DOAC Filter^®^ treatments in a limited sample of four patients who were not on DOAC treatment. Other authors have studied the effects of DOAC removal on other coagulation tests in more detail [10,16,18,20,24,25,26,27].

## 5. Conclusions

In conclusion, DOAC Filter^®^ and DOAC-Stop™ are effective in removing DOAC from patient samples ex vivo, and DOAC removal is associated with reduced frequency of LAC screening test positivity, especially in RVVT-based tests. However, 46% of patient samples had positive LAC screening test results even after DOAC removal. Some of these patients may harbor a true LAC. Repeated testing would be required if LAC testing results remain positive after DOAC removal. However, published results and our data show that a significant proportion of patients with a positive LAC screening test would not require repeated LAC testing if the samples were processed with a DOAC removal protocol. This could reduce healthcare-related costs and help in making correct and early diagnoses for thrombophilia patients.

## Figures and Tables

**Figure 1 diagnostics-12-02520-f001:**
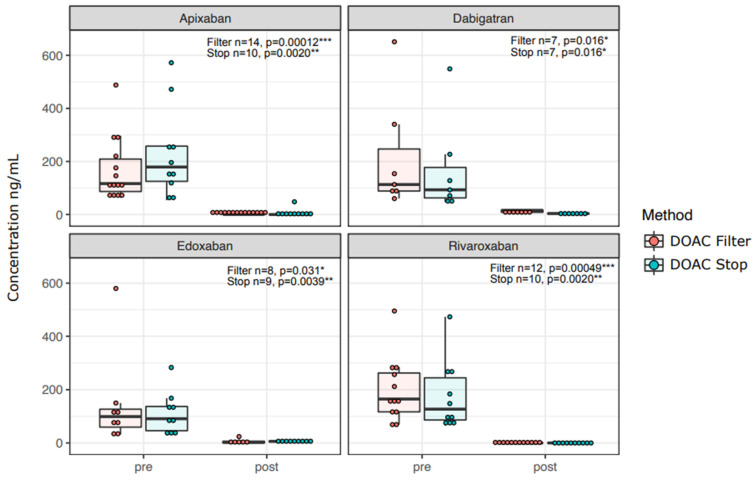
DOAC concentrations are effectively reduced by both DOAC Filter^®^ and DOAC-Stop™. Concentrations of DOACs (as ng/mL) before (pre) and after (post) DOAC removal via DOAC Filter^®^ or DOAC-Stop™ are shown. The paired sign test was used to test for significance. Asterisks: *: *p* < 0.05; **: *p* < 0.01, ***: *p* < 0.001

**Figure 2 diagnostics-12-02520-f002:**
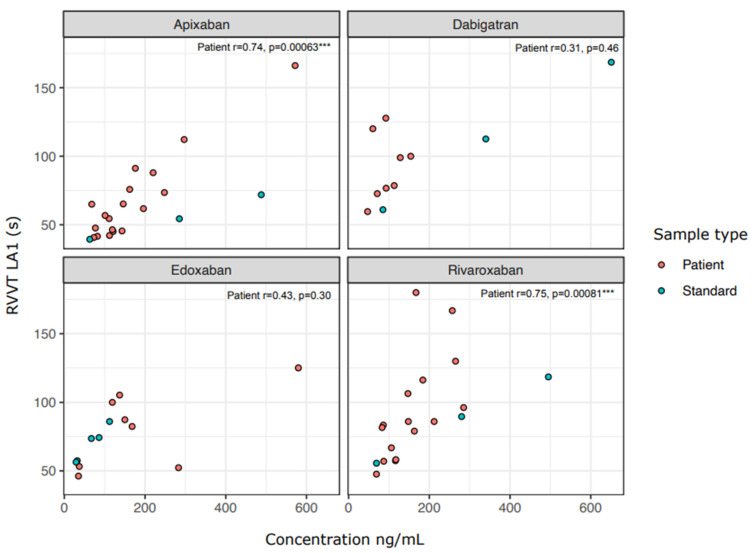
Lupus anticoagulant RVVT test screening assay results correlate with DOAC concentration. Results for the RVVT-based LA1 assay are shown prior treatment with DOAC-Stop™ or DOAC Filter^®^. Correlations were calculated using Spearman correlation and the *p*-value and Spearman rho are shown in the panels. Abbreviations: api, apixaban; dabi, dabigatran; edo, edoxaban; riva, rivaroxaban; conc, concentration as ng/mL. Asterisks: ***: *p* < 0.001.

**Figure 3 diagnostics-12-02520-f003:**
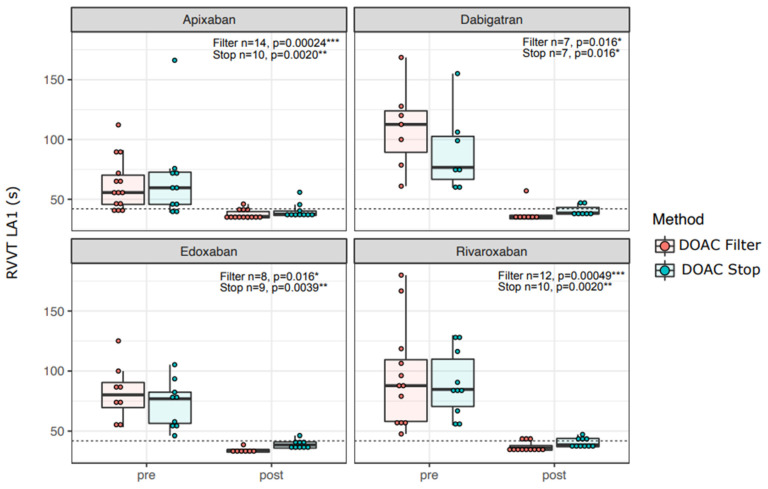
RVVT screening test clotting times reduce with DOAC removal. RVVT- test screening test (LA1) clotting times before and after DOAC Filter^®^ and Stop™ treatments. The paired sign test was used to test for statistical significance. The dotted line represents the cutoff for a positive screening test (42 s). Asterisks: *: *p* < 0.05; **: *p* < 0.01, ***: *p* < 0.001

**Figure 4 diagnostics-12-02520-f004:**
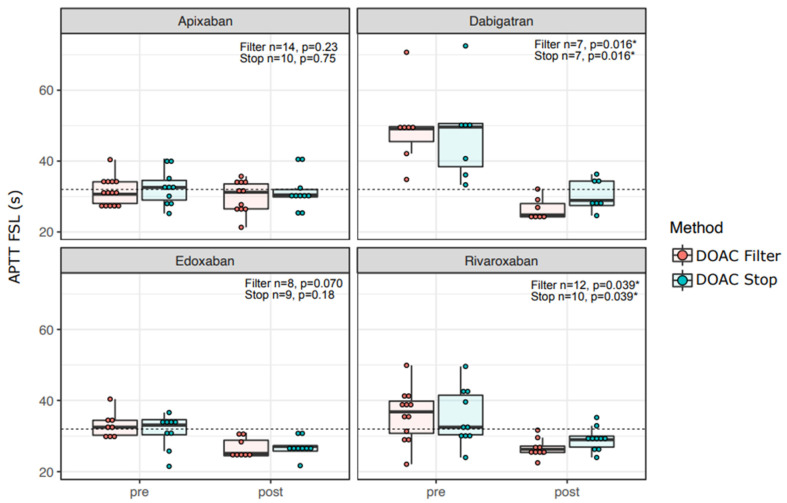
APTT-based screening test clotting times reduce only for samples with dabigatran or rivaroxaban when treated with DOAC adsorbents. APTT-based test screening test (FSL) clotting times before and after DOAC Filter^®^ and Stop™ treatments. The paired sign test was used to test for statistical significance, and *p*-values are shown in the figure. The dotted line represents the cutoff for a positive screening test (32 s). Asterisk: *: *p* < 0.05.

**Figure 5 diagnostics-12-02520-f005:**
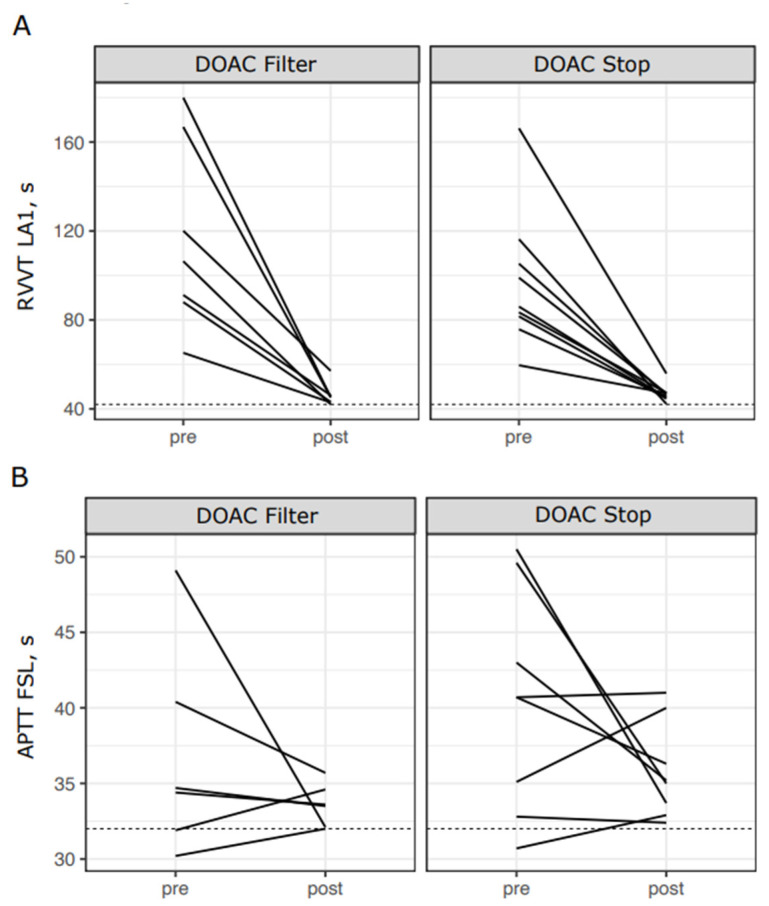
DOAC removal methods reduce clotting times even though the screening test results remain positive. The figure shows results for samples in which the RVVT screening test results ((**A**), LA1) or APTT-based screening test ((**B**), APTT.FSL) remained positive even after DOAC removal by DOAC Filter^®^ or DOAC-Stop™. The median difference for LA1 was −63 s for DOAC Filter^®^-treated samples (*p* = 0.016) and −41 s for DOAC-Stop™-treated samples (*p* = 0.0039). The median difference for APTT.FSL was −1.0 for DOAC Filter^®^-treated samples (*p* = 0.69) and −2.4 for DOAC-Stop-treated samples (*p* = 0.73). Abbreviations: pre, before DOAC removal; post, after DOAC removal.

**Table 1 diagnostics-12-02520-t001:** DOAC concentrations and Lupus anticoagulant results before and after DOAC removal.

			Before DOAC Removal	After DOAC Removal	Before DOAC Removal	After DOAC Removal
	DOAC in Sample	Samples (n)	Concentration Median (Range), ng/mL	Concentration Median (Range), ng/mL	Positive Lupus Anticoagulant Screening	
Patient samples: DOAC Filter^®^	Apixaban	11	112 (68–297)	0 (0–15)	10/11	7/11
	Dabigatran	4	103 (60–154)	9 (0–18)	4/4	1/4
	Edoxaban	4	134.5 (37–580)	6 (5–7)	4/4	0/4
	Rivaroxaban	9	163 (69–285)	0 (0–4)	9/9	3/9
Patient samples: DOAC-Stop™	Apixaban	7	162 (73–572)	0 (0–48)	6/7	4/7
	Dabigatran	4	82 (47–128)	4 (2–5)	4/4	3/4
	Edoxaban	4	152 (35–283)	7 (6–8)	4/4	1/4
	Rivaroxaban	7	106 (83–265)	0 (0–0)	7/7	4/7
Calibrators and controls: DOAC Filter^®^	Apixaban	3	285 (63–488)	3 (0–6)	2/3	0/3
	Dabigatran	3	340 (85–651)	16 (16–17)	3/3	0/3
	Edoxaban	4	76.5 (32–112)	0 (0–24)	4/4	0/4
	Rivaroxaban	3	280 (69–495)	0 (0–0)	3/3	0/3
Calibrators and controls: DOAC-Stop™	Apixaban	3	261 (54–472)	0 (0–0)	2/3	0/3
	Dabigatran	3	227 (54–549)	3 (2–4)	3/3	0/3
	Edoxaban	5	78 (29–131)	5 (5–7)	5/5	0/5
	Rivaroxaban	3	271 (65–473)	0 (0–0)	3/3	0/3

Abbreviations: DOAC, direct oral anticoagulant.

## Data Availability

Data are available from the corresponding author on request.

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
