# Peer review of "Novel Ex Vivo DOAC Removal Methods Reduce Interference in Lupus Anticoagulant Testing"

_diagnostics, 2022, doi:10.3390/diagnostics12102520_

Round 1

Reviewer 1 Report

This small study deals with the clinically relevant issue of the interaction of direct oral anticoagulants (DOAC) in plasma-based assays, in particular when lupus anticoagulant (LAC) is determined. 

The paper is well written, and I do not have any issues related to language or style. 

A few comments:

1.     Indeed, this concerns “real life” data from leftover plasma samples, which is different from spiked samples and overall the number of tested samples is acceptable, given the huge effects of DOAC remove, but to address the impact of removal agents on routine clotting tests, requires more than 4 samples. Others also addressed this, eg Monteyne et al, Thromb Res 2020. Preferably, I would remove the data from table 2 from this paper or expand it to allow a reasonable comparison of pre and post removal effects. 

2.     I like the data in fig 1,4,5 and 6 and table 1; data from fig 2 and 3 seem less relevant and I wonder how important these illustrations are to convey the main message? 

3.     Altogether, this work shows how important it is to be informed in the laboratory about the possible presence of a DOAC in a clinical sample. In our lab, the policy is not to report possible false positive data in presence of a DOAC and to verify with the clinic whether a definitive answer on LAC is required. Next, a DOAC removal test is added. My question: in case it is essentially unknown whether a DOAC is in the plasma, do you advocate the routine application of a DOAC remove to verify? This would be an important consequence, I think. 

4.     For this small study, the discussion is quite long; please try to condense.  

Author Response

  1. Thank you for pointing this out. We agree that data on four patients is not sufficient to draw conclusive remarks on the effects of DOAC Stop or DOAC Filter on other coagulation tests. We removed the table 2 from the main manuscript to the supplementary material. We also appended the Results -section (3.1) text to emphasize that our studies included only a very small number of samples: “We also analyzed a limited sample of four patient samples without DOAC therapy to assess if DOAC-Stop™ or DOAC Filter® had effects on coagulation tests” (row 141 of the revised manuscript with markings). The discussion on the effects of DOAC removal on other coagulation tests than LAC was abbreviated and the reader is referenced to other, more comprehensive studies on this subject, including now also Monteye et al: “Other authors have studied the effects of DOAC removal in more detail [10,16,18,20,24–27]” (rows 332-333 of the revised manuscript with markings).
  2. We replaced the Figure 3 in the Supplementary Material, but left Figure 2, as correlations in the RVVT assay were more pronounced, in the main manuscript.
  3. In our clinical lab, we receive structured clinical information on the DOAC used in the laboratory referral. Thus, in many cases, we have reasonably accurate information on the DOAC used. However, in the future we plan to use DOAC Stop also on patient samples that lack information on whether the patient is using DOAC if the LAC testing is positive. We also added emphasis on the importance of this information in the discussion of the revised manuscript: “Thus, ensuring that the laboratory referral is accurate on whether DOAC is used is of primary importance”(rows 314-315 of the revised manuscript with markings)
  4. Thank you for the feedback. We have shortened the discussion, and we especially abbreviated the section discussing DOAC Stop/DOAC Filter effects on other coagulation tests because our manuscript does not have extensive data on that subject.

Reviewer 2 Report

I find this study as interesting and beneficial for the patients with obtaining the objective results of LAC testing.

However, I would like to kindly ask the authors to add more data about some clinical feaures of the patients, if available. Thus, I recommend minor revision of the manuscript.

Comments to the authors:

Content suggestions:

1. I would like to kindly ask the authors to reveal the following data about the patients: age, sex, comorbidities with the focus on renal or hepatic impairment (knowledge about the need to modify the dose of DOACs due to CrCl or liver function tests).

2. Did these patients have any adverse effects of the treatment (allergic reaction, bleeding events, thromboembolic episode, etc. ) ?

3. Did the authors include the patients with primary or also secondary LAC positivity ?

Thus, I recommend minor revision of the manuscript.

Author Response

  1. Thank you for asking this. We believe that additional clinical information on the patients would have been interesting, but unfortunately we do not have access to clinical information of these patients: sample collection was anonymous, and our laboratory has only information on the laboratory test ordered and information on the DOAC the patient may have been using. Thus, we regret that we cannot add additional clinical information on these patients.
  2. Because we do not have access to detailed clinical information of these patients, unfortunately we do not know if any of the study patients had bleeding events or other adverse events during the DOAC treatment.
  3. We selected patient samples based on suspicion of DOAC interference: samples with higher DOAC concentrations were preferred in selection. This is now further explained in the manuscript results in the beginning of section 3.3: “The patient samples in this study were selected based on suspicion of DOAC interference”. Although secondary LAC was suspected in most cases, LAC screening remained positive even after DOAC Stop or DOAC Filter treatment in 46% of patient samples. Real, primary LAC or factor deficiency cannot be ruled out because we did not have sufficient amount of sample to perform mixing studies for patient samples in this study.